# Synthesis of a covalent organic framework with hetero-environmental pores and its medicine co-delivery application

Wenyan Ji[1,2,7], Pai Zhang [3,7], Guangyuan Feng[2,7], Yuan-Zhe Cheng [1,4], Tian-Xiong Wang[1,4], Daqiang Yuan [5], Ruitao Cha [3] ✉, Xuesong Ding[1] ✉, Shengbin Lei [2,6] ✉ & Bao-Hang Han [1,4] ✉

The topology type and the functionalization of pores play an important role in regulating the performance of covalent organic frameworks. Herein, we designed and synthesized the covalent organic framework with hetero-environmental pores using predesigned asymmetrical dialdehyde monomer. According to the results of structural characterization, crystallinity investigation, and theoretical calculation, the hetero-environmental pores of the obtained framework are regarded as the alternant arrangement. The distinctive hetero pore structure leads the designed material to show more advantages as compared with control materials in loading both hydrophobic and hydrophilic antibiotics for wound healing. This dual-antibiotic strategy can expand the antibacterial range as compared with the single antibiotic one, and reduce the generation of drug resistance. In summary, this strategy for designing covalent organic frameworks with hetero-environmental pores can extend the structural variety and provide a pathway for improving the practical application performance of these materials.

Covalent organic frameworks (COFs) have become a powerful platform for the design of functional materials since the pioneering work has been reported by Yaghi and co-workers in 2005[1]. COFs can be constructed by integrating components into two-dimensional (2D) polymers to form ordered nanopores, which are designable, adjustable, and modifiable through pore engineering[2–5]. Meanwhile, COFs possess precise spatial structures and regular block distributions, which result in forming periodic columnar π-arrays and directional one-dimensional (1D) channels[6,7]. The structural research on COFs mainly focuses on the topology of the framework and the surface engineering of the channels[8,9]. The different topological structures of the frameworks generate various shapes and sizes of the pores, which can be obtained by elaborately designing the building blocks with different sizes and specific symmetries[10–15]. The channel surface engineering of COFs is mainly concentrated on regulating the chemical composition of the framework and the functionalization of the pores through the process of pre-synthesis modification or post-synthesis functionalization[16–19]. Since Zhao and co-workers reported a dual-porous COF with the periodic and orderly distribution of micropores and mesopores in 2014, a series of COFs with multiple pores in one framework have been prepared[20]. These reported heteroporous COFs exhibit some specific interesting properties[21–24]. For example, Ma and

[1]CAS Key Laboratory of Nanosystem and Hierarchical Fabrication, CAS Center for Excellence in Nanoscience, National Center for Nanoscience and Technology, Beijing 100190, China. [2]Department of Chemistry, School of Science & Collaborative Innovation Center of Chemical Science and Engineering (Tianjin), Tianjin University, Tianjin 300072, China. [3]CAS Key Laboratory for Biomedical Effects of Nanomaterials and Nanosafety, CAS Center for Excellence in Nanoscience, National Center for Nanoscience and Technology, Beijing 100190, China. [4]University of Chinese Academy of Sciences, Beijing 100049, China. [5]State Key Laboratory of Structure Chemistry, Fujian Institute of Research on the Structure of Matter, Chinese Academy of Sciences (CAS), Fuzhou 350002, China. [6]Tianjin Key Laboratory of Molecular Optoelectronic Science, Tianjin University, Tianjin 300072, China. [7]These authors contributed equally: Wenyan Ji, Pai Zhang, Guangyuan Feng. ✉e-mail: chart@nanoctr.cn; dingxs@nanoctr.cn; shengbin.lei@tju.edu.cn; hanbh@nanoctr.cn

co-workers proved that dual-porous COFs are more advantageous than single-porous COFs in the enzyme host-catalyzed organic reaction[25]. In their design, the large hexagonal pores are employed to encapsulate enzymes as a catalytic space, while the small triangular holes are used as channels for transporting reagents and products. The coexistence of pores with multiple sizes can effectively minimize diffusion barriers. The heteroporous COFs with various pore shapes and sizes have been well-studied, however, the chemical environment of the pore channels is homogeneous. The COFs with different chemical environments of pore channels are seldom investigated.

Antibiotic-resistant bacteria pose a significant threat to human health. The development of antibiotics lags behind the evolution of bacterial resistance, highlighting the urgent need to effectively utilize existing antibiotics. Aminoglycoside/beta-lactam combinations exhibit synergistic effects and are the most commonly used combinations in clinical practice[26]. By combining antibiotics with different/opposite properties can overcome resistance mechanisms of bacteria and increase the treatment effectiveness[27]. For complex infections caused by drug-resistant bacteria, the combination of antibiotics is an emerging and important strategy to effectively combat drug-resistant bacterial infections. However, the compatibility of different drugs with disparate properties (such as the different hydrophilicity) limits the combination of drugs[28,29]. In recent years, various antibacterial materials and strategies have been developed[30]. Shi et al. incorporated hydrophilic (−)-epigallocatechin-3-O-gallate (EGCG) in the shell of pH-responsive mixed shell polymeric micelle, and loaded hydrophobic ciprofloxacin in the core. It is proved that the co-delivery system of hydrophilic/hydrophobic drugs possesses balanced biofilm dispersal and killing[31]. Fan et al. redesigned the inner surface of ferritin drug carrier (ins-FDC) nanocarrier as a drug delivery platform and used the urea-thermal incubation loading method to maximize the loading capacity of hydrophilic and hydrophobic drugs at the same time. The result shows that the dual-drug loaded system exhibits a higher synergistic antitumor effect than the single-drug loaded ones and efficiently overcomes the tumor cell drug resistance[32]. The co-delivery system is imperative and should be further explored. Previous studies demonstrate that COFs can serve as carriers to upload and deliver drugs for therapeutic applications[33,34]. If the designed COF contains unique hetero-environmental pores, it can be predicted to be an ideal co-delivery carrier for drugs.

In this work, we designed and synthesized an imine-linked COF, DEG-HEP-COF, by connecting 5,10,15,20-tetrakis(4-aminophenyl)porphyrin (TAPP) and dialdehyde monomer with both hydrophobic alkyl chain and hydrophilic alkoxy chain in one molecule. Meanwhile, the COFs containing only hydrophilic chains (DEG-COF) or hydrophobic chains (HEP-COF) were constructed and utilized as control samples to evaluate the structures and properties of DEG-HEP-COF. Furthermore, in order to investigate the effect of the uniformity of pore distribution on the material properties, another control sample (DEG+HEP-COF) was prepared via the multi-component copolymerization method, which possesses the random and heterogeneous porous channels of hydrophoblity and hydrophility. Powder X-ray diffraction (PXRD) measurement, nitrogen adsorption–desorption experiment, and simulation calculation results indicate that DEG-HEP-COF possesses hetero-environmental pores, which are arranged in periodic arrays similar to the chess board. Based on its unique hetero-environmental pores, the DEG-HEP-COF was used to load hydrophobic levofloxacin (Lev) and hydrophilic vancomycin (Van) to explore the relationship between the pore environment and the loading/relasing behaviors of antibiotics. The experimental results show that the dual-antibiotic delivery strategy by using the DEG-HEP-COF carrier can expand the antibacterial range as compared to a single antibiotic, and delay or reduce the generation of drug resistance. Therefore, this kind of COFs containing hetero-environmental pores can be similarly prepared by designing monomers elaborately, and this synthesis strategy can expand the structural and functional variety of COF materials.

## Results

### Synthesis and characterizations of the chemicals

The monomers have been synthesized and characterized by the Fourier transform infrared (FT-IR) spectroscopy, $^1H$ NMR, $^{13}C$ NMR, and high-resolution mass spectra (HR-MS) (Supplementary Fig. 1 and Table 1). The DEG-HEP-COF with regular distributed alkoxy chain (diethylene glycol, DEG) and alkyl chain (heptyl, HEP) in the channel walls was prepared by the condensation reaction (Fig. 1) of TAPP (Supplementary Figs. 2 and 3) and DEG-HEP-CHO (Supplementary Figs. 4–7). In order to facilitate comparison of the DEG-HEP-COF, we designed and synthesized the corresponding control materials (Fig. 1), including DEG-COF by the reaction of TAPP and DEG-CHO (with only alkoxy chains) (Supplementary Figs. 8–11), HEP-COF by combining TAPP with HEP-CHO containing only alkyl chains (Supplementary Figs. 12–15), and DEG+HEP-COF with randomly distributed alkyl and alkoxy chains via copolymerization of TAPP, DEG-CHO, and HEP-CHO (Fig. 1). Firstly, the optimized synthesis conditions were screened by changing the reaction solvent (type and ratio), temperature, and reaction time (Supplementary Tables 2–5), which were assessed by PXRD measurement (Supplementary Figs. 16–19). The experiment results show that the type and ratio of reaction solvents make a huge impact on the crystallinity of the yielded materials. The DEG-HEP-COFs with optimized crystallinity and Brunauer–Emmett–Teller (BET) specific surface area can be obtained under several different conditions, including a mixture of $o$-dichlorobenzene/ethanol/acetic acid (6 M) (1/9/2 in volume ratio), 1,4-dioxane/ethanol/acetic acid (6 M) (1/9/2 in volume ratio), or mesitylene/ethanol/acetic acid (6 M) (1/9/2 in volume ratio) at 120 °C for 3 d. The variety of optimized reaction conditions results from the adaptability of the amphiphilic monomer to different solvents, which reduced the difficulty in synthesis condition screening. Here, the mixture of $o$-dichlorobenzene/ethanol/acetic acid (6 M) (1/9/2 in volume ratio) was chosen as the reaction condition of DEG-HEP-COFs that was used to investigate the subsequent structure and performance. The control COFs were obtained under the optimized preparation conditions of ethanol/acetic acid (6 M) (5/1 in volume ratio) for DEG-COF, $o$-dichlorobenzene/ethanol/acetic acid (6 M) (7/3/2 in volume ratio) for HEP-COF, and $o$-dichlorobenzene/ethanol/acetic acid (6 M) (5/5/2 in volume ratio) for DEG+HEP-COF at 120 °C for 3 d, respectively. Then, the FT-IR, solid-state $^{13}C$ NMR measurement, X-ray photoelectron spectroscopy (XPS), and elemental analysis were utilized to confirm the structures of these COFs. FT-IR spectra show stretching vibration peaks at 1601 cm$^{-1}$ for all these COFs, which can be assigned to the C=N bond and demonstrates the formation of imine linkages (Supplementary Fig. 20). The stretching bands of N–H and C=O bonds in the spectra of the corresponding monomers at 3320 and 1690 cm$^{-1}$ almost disappeared in the spectra of these COFs (except DEG+HEP-COF), suggesting the high imine condensation degree. The phenomenon in DEG+HEP-COF is owing to the reactivity difference between DEG-CHO and HEP-CHO, which may cause incomplete reaction of the terminal aldehyde groups. Meanwhile, several peaks in the range of 2850–3000 cm$^{-1}$ can be assigned to the stretching vibration band of –CH$_3$ and –CH$_2$ belonging to the alkyl groups and alkoxy groups in the COF skeletons. The solid-state $^{13}C$ NMR spectra of these COFs exhibit the characteristic signal at 158 ppm for carbon in C=N bond, and the characteristic signals at 152, 149, 140, 132, 129, 119, and 115 ppm corresponding to the aromatic carbons in benzene ring and the carbons in porphyrin. The alkoxy carbon (–C–O–) and alkyl carbon (–C–C–) are observed at 56–72 and 11–31 ppm, respectively (Supplementary Fig. 21). The XPS results are also consistent with the chemical structure of COFs (Supplementary Fig. 22). Furthermore, the elemental analysis data show that the C, H, and N contents of the four COFs are in

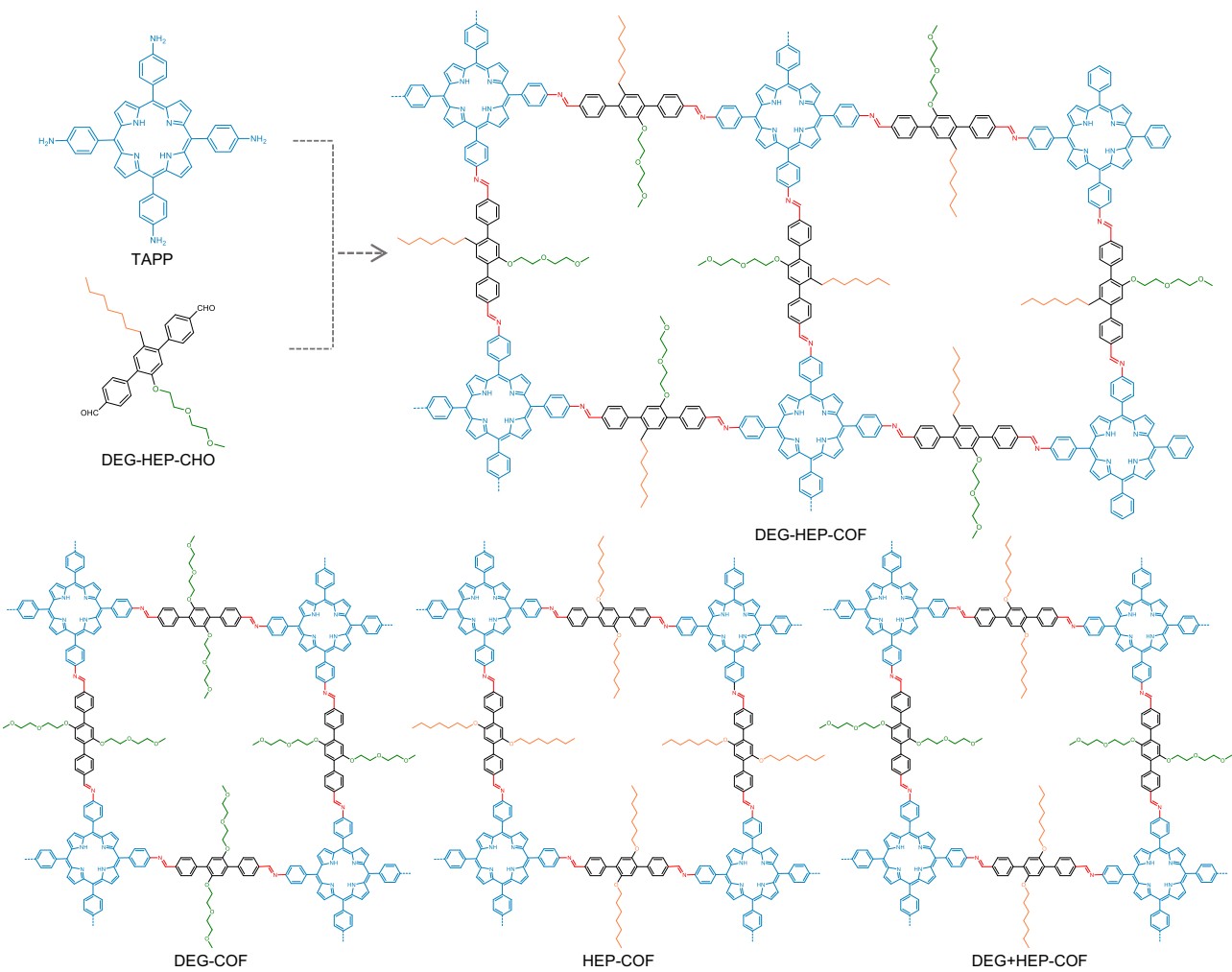

**Fig. 1 | Design and synthesis of the COF materials.** Schematic illustration of the synthesis of DEG-HEP-COF with hetero-environmental pores, and the structures of control materials, including DEG-COF, HEP-COF, and DEG+HEP-COF.

agreement with the theoretical calculation values (Supplementary Table 6).

## Porositiy and stability

The porosity of these COFs was investigated by the nitrogen adsorption–desorption isotherm measurement at 77 K. As shown in Fig. 2a and Supplementary Fig. 23a, the sorption curves of DEG-HEP-COF and DEG-COF show typical type I isotherm[35], and possess BET specific surface area of 1030 and 920 m$^2$ g$^{-1}$, respectively. The total pore volumes of DEG-HEP-COF and DEG-COF are calculated to be 0.49 and 0.50 cm$^3$ g$^{-1}$, respectively, on the basis of the single point adsorption at $P/P_0 = 0.97$. The pore sizes of these COFs center at 1.4 nm by the calculation of Original Density-Functional Theory, suggesting that the two samples are micropore-dominated materials. As shown in Supplementary Fig. 23b, the sorption isotherm of HEP-COF exhibits low BET specific surface area of 20 m$^2$ g$^{-1}$ and low pore volume of 0.03 cm$^3$ g$^{-1}$. The sorption curves of DEG+HEP-COF shows the combination of type I and II isotherm[35], with BET specific surface area of 280 m$^2$ g$^{-1}$, pore volume of 0.18 cm$^3$ g$^{-1}$, and the main pore diameter of 1.4 nm (Supplementary Fig. 23c). Thermal stabilities of these COFs were measured by thermogravimetric analysis (TGA) under N$_2$ atmosphere (Supplementary Fig. 24). All these COFs do not show obvious weight loss even until 400 °C, revealing their excellent thermal stability. Meanwhile, the chemical stability of DEG-HEP-COF was investigated. As shown in Supplementary Fig. 25, the crystallinity can be completely retained when the sample is soaked in tetrahydrofuran (THF),

dimethylformamide (DMF), dimethyl sulfoxide (DMSO), methanol (MeOH), or *n*-hexane for 7 d, respectively. The chemical structure of the soaked sample was also examined by FT-IR, which exhibits an ignorable variation as compared with the original COFs. Consequently, the DEG-HEP-COF possesses strong chemical stability in organic solvents. However, COFs are unstable in aqueous HCl (1 M) and NaOH (1 M) solutions, obviously owing to the disassociation of imine linkages (Supplementary Fig. 26).

## Crystallinity and stacking structure

The crystallinity of these COFs was evaluated by PXRD measurement. As shown in Fig. 2b, the DEG-HEP-COF exhibits a strong and sharp diffraction peak at around $2\theta = 3.2°$ corresponding to the (110) facet. Other diffraction peaks in the PXRD pattern of DEG-HEP-COF at around 3.7°, 4.7°, 5.4°, 6.1°, 6.5°, 7.3°, and 7.9° are assignable to (020), (120), (200), (210), (220), (001), and (111) facets, respectively. The lattice mode of DEG-HEP-COF was simulated and the geometric energy of structures was minimized by the Material Studio. We noticed that the experimental PXRD pattern could not completely agree with the simulated pattern of conventional eclipsed AA or staggered AB stacking mode (Supplementary Fig. 27). There is a deviation between the main diffraction peak of simulated conventional eclipsed AA stacking mode ($2\theta = 2.8°$) and the experimental one (3.2°). According to the reported similar structure[36–38], we allow the lattice distortion from regular square nets to the optimized conformation on the basis of the eclipsed AA stacking. As shown in Fig. 2c, simulation using the

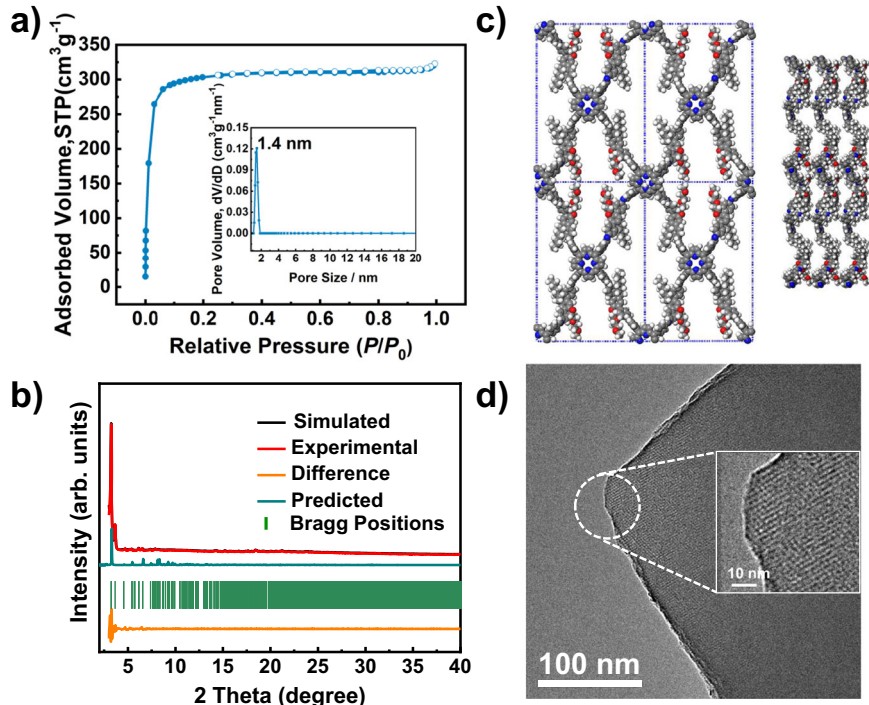

**Fig. 2 | Porous properties, crystallinity and stacking structure. a** $N_2$ adsorption–desorption isotherm and pore size distribution profile (inset) of DEG-HEP-COF. **b** PXRD profiles of DEG-HEP-COF of the experimentally observed (red), Pawley refined (black), their difference (orange), simulated (cyan) using the contorted AA stacking mode, and the observed reflections (green). **c** Side and top view of the contorted eclipsed (AA) structure of DEG-HEP-COF (gray: C, blue: N, red: O, white: H). **d** HR-TEM image of DEG-HEP-COF. (The three independent samples were recorded, and 1 representative image is shown). Source data are provided as a Source Data file.

contorted AA stacking mode well matches the experimental PXRD pattern of DEG-HEP-COF with a negligible difference, and the Pawley refinement (space group with the cell unit parameters of $a = 32.4$ Å, $b = 47.6$ Å, $c = 11.9$ Å; $\alpha = \beta = \gamma = 90°$) (Supplementary Table 7,) shows good agreement factors with experimental data ($R_{wp} = 6.66\%$, $R_p = 3.64\%$). In terms of the structure, the phenyl of the porphyrin and the terphenyl are twisted in contorted AA stacking mode. This phenomenon has been observed in other reported works about similar structures[37]. The steric hindrance of the chains of the adjacent layers prevents the formation of framework with a close-packed face-on orientation, and causes the formation of framework with a less closely twisted stacked conformation[39]. The flexibility of terphenyl moiety and the mutual attraction of side chains endow the bending and stacking of the skeleton with balance, which guarantee crystallinity of the COFs under the contorted stacking. The high-resolution transmission electron microscopy (HR-TEM) image shows highly ordered lattice fringes, confirming the periodically ordered pore structure of DEG-HEP-COF (Fig. 2d). In the meanwhile, the DEG-COF also exhibits a sharp diffraction peak at around $2\theta = 3.2°$ in PXRD pattern, which shows the analogous diffraction peaks and lattice mode in structural simulation (space group with the cell unit parameters of $a = 32.4$ Å, $b = 47.6$ Å, $c = 11.9$ Å; $\alpha = \beta = \gamma = 90°$, $R_{wp} = 14.74\%$, $R_p = 11.43\%$) to those of DEG-HEP-COF (Supplementary Fig. 28 and Table 8). The HR-TEM image of DEG-COF also shows a highly ordered arrangement (Supplementary Fig. 29). While HEP-COF presents weak and broad diffraction peaks at 5.9° and 20.7°. Interestingly, as compared with DEG-HEP-COF, DEG+HEP-COF with random distribution of alkoxy chains and alkyl chains shows weaker peaks at 3.4°, 6.7°, 7.4°, and 23.2° corresponding to the (110), (220), (001), and (111) facets. Over all, the difference of the alkyl chain and alkoxy chain in monomers results in a great difference in porosity and crystallinity of COFs. In addition, the electrostatic potential distribution of monomers was calculated. The aldehyde groups on DEG-HEP-CHO showed more negative than those on DEG-CHO, and more positive than those on HEP-CHO (Supplementary Fig. 30). Thus, the amphiphilic DEG-HEP-CHO combining the alkyl chains and alkoxy chains exhibits a moderate reaction rate[40,41], which leads to the DEG-HEP-COF with ordered pore structure, uniform pore diameter distribution, and high BET specific surface area[42,43], The field-emission scanning electron microscopy (FE-SEM) measurement reveals significant differences in the morphology of these COFs (Supplementary Fig. 31). Considered that the differences of these COFs may originate from their structure, hence the growth process of these COFs is carried out to explore the relationship between their structures and those differences in porosity and crystallinity.

## Growth process of COFs

The formation process of these COFs was analyzed by the changes in crystallinity, porosity, and morphology using PXRD, nitrogen sorption isotherm, and FE-SEM measurements, respectively. The diffraction peaks of DEG-HEP-COF cannot be observed at 1 d in the PXRD pattern, which indicates DEG-HEP-COF is still in amorphous state (Fig. 3a). As compared with the DEG-HEP-COF, the diffraction peaks of DEG-COF can be observed in a shorter reaction time (0.5 d), which means that this material possesses a faster polymerization or crystallization rate as compared with that of DEG-HEP-COF (Fig. 3b). When the reaction time is extended to 2 d, the DEG-HEP-COF shows a certain extent of crystallinity with a crystal length of about 4.0 µm (Fig. 3a and Supplementary Fig. 32a). After continuous reaction for another 24 h, and the DEG-HEP-COF crystallinity apparently increases with crystal length of about 5.0 µm. As compared with the DEG-HEP-COF, the particle size of DEG-COF is about 2 µm after 3 d (Supplementary Fig. 32b). Significantly, when the reaction time is prolonged to 14 d, the crystallite size of the DEG-HEP-COF increases to 10 µm, and the shape and morphology become much uniform (Fig. 3c). The variation tendencies on the BET specific surface area of the DEG-HEP-COFs and DEG-COFs are

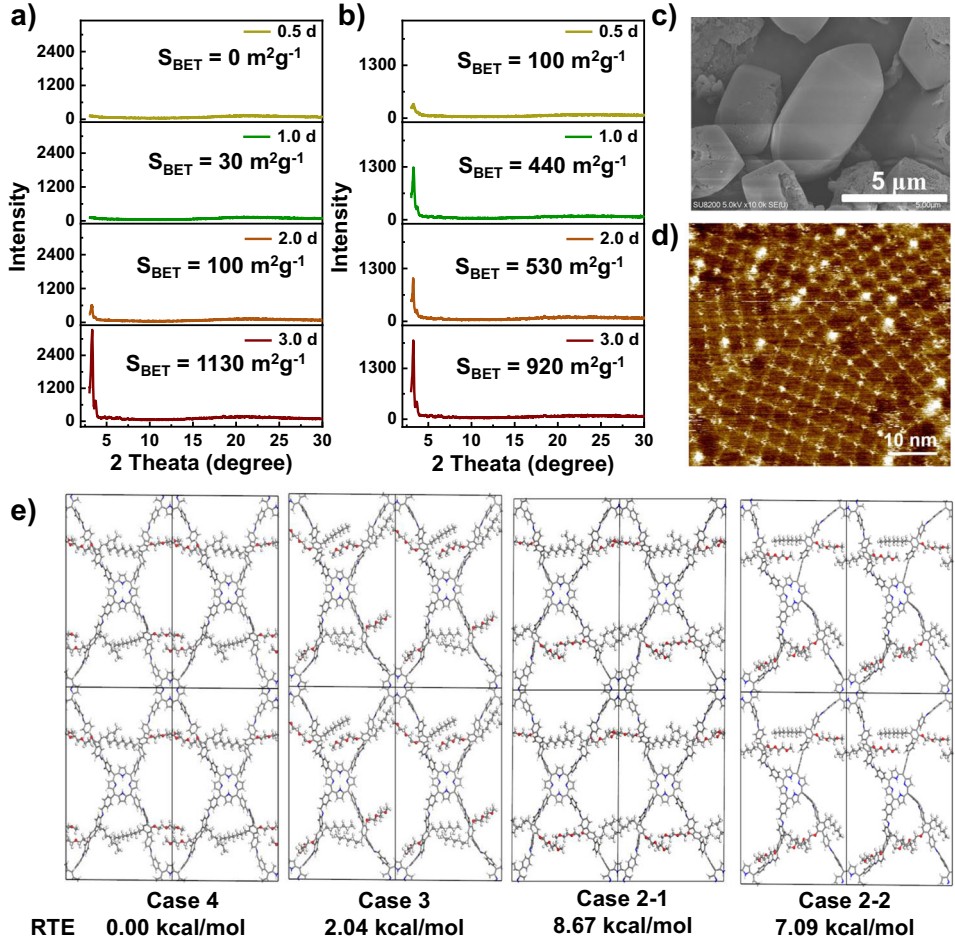

**Fig. 3 | Growth process and the pore structure verification of COFs. a** BET specific surface area values and PXRD patterns of DEG-HEP-COF recorded at different reaction time. **b** BET specific surface area values and PXRD patterns of DEG-COF recorded at different reaction time. **c** SEM image of DEG-HEP-COF recorded at 14 d. **d** STM image of DEG-HEP-COF. STM image obtained by co-condensation of TAPP and DEG-HEP-CHO with thermal annealing in water atmosphere. Imaging parameters: $I_{set}$ = 60 pA, $V_{bias}$ = −0.6 V. **e** The RTE of tetragons with different chain orientations. Case-4 shows four alkyl chains in the same hole. Case-3 shows three alkyl chains in the same hole. Case-2-1 and Case-2-2 shows two alkyl chains in the same hole, which are in ortho-position and in para-position, respectively. (For **c, d**, three independent samples were recorded, and 1 representative image is shown. The high-resolution version of **e** is included in the Supplementary information, Supplementary Fig. 38). Source data are provided as a Source Data file.

analogous to those on PXRD data. The BET specific surface area of DEG-HEP-COF increases obviously after reaction more than 2 d, which is from less than 100 to 1030 m² g⁻¹. However, the BET specific surface area values of DEG-COF are 100, 440, 530, and 920 m² g⁻¹ at 0.5, 1.0, 2.0, and 3.0 d, respectively, which indicate rapid growth at the beginning of DEG-COF preparation. For the case of HEP-COF and DEG+HEP-COF, the obvious phenomenon of crystallite growth cannot be observed in the whole reaction process. Meanwhile, the BET specific surface area values and intensities of the diffraction peaks in PXRD patterns of HEP-COF and DEG+HEP-COF are relatively weak during the whole reaction process, indicating the effective interlayer stacking process are not realized (Supplementary Table 9 and Figs. 33–36). The obvious differences in the growth process between DEG-HEP-COF and other control COFs originate from their structural differences (vide infra). Furthermore, the scanning tunneling microscopy (STM) with atomic resolution was employed to tentatively study the arrangement of the two side chain groups. As shown in Fig. 3d, the co-condensation of TAPP and DEG-HEP-CHO was performed with thermal annealing in water atmosphere to gain the single-layer atomically precise nanostructures. In STM results, the TAPP appears as a bright feature, from this point, we could determine the location of TAPP and the structure of surface DEG-HEP-COF. However, due to the resolution and

technical features, we cannot directly observe the side chain arrangement of DEG-HEP-COF. Hence the theoretical calculations are carried out to further explore the structure of DEG-HEP-COF.

## Hetero-environmental pore structure
We firstly calculated the interlayer interaction energy of the DEG/ HEP chains of COFs' fragments. The fragment of DEG-COF presents the most favorable interaction energy among these three samples (Supplementary Fig. 37), which is beneficial to the stacking structure of COFs to promise the crystallinity. It can be speculated that the alkoxy chain is conducive to crystallization, which may be analogy to the crystallization behavior of polyethylene glycol that has been studied in previous researches[44]. This result shows good agreement with the phenomenon in above-mentioned formation process study that the DEG-COF containing hydrophilic ethylene oxide chains exhibits a fast crystallization. In contrast, the fragment of HEP-COF presents the weakest interaction energy among these three samples, and the low crystallinity of HEP-COF probably originates from that the alkyl chains severely affect the stacking and crystallizing process[45]. From the structural perspective, the DEG-HEP-COF containing half amounts of the alkyl chains may exhibit weakened crystallinity to a certain extent. However, the DEG-HEP-COF possessing good crystallinity indicated that the interaction of

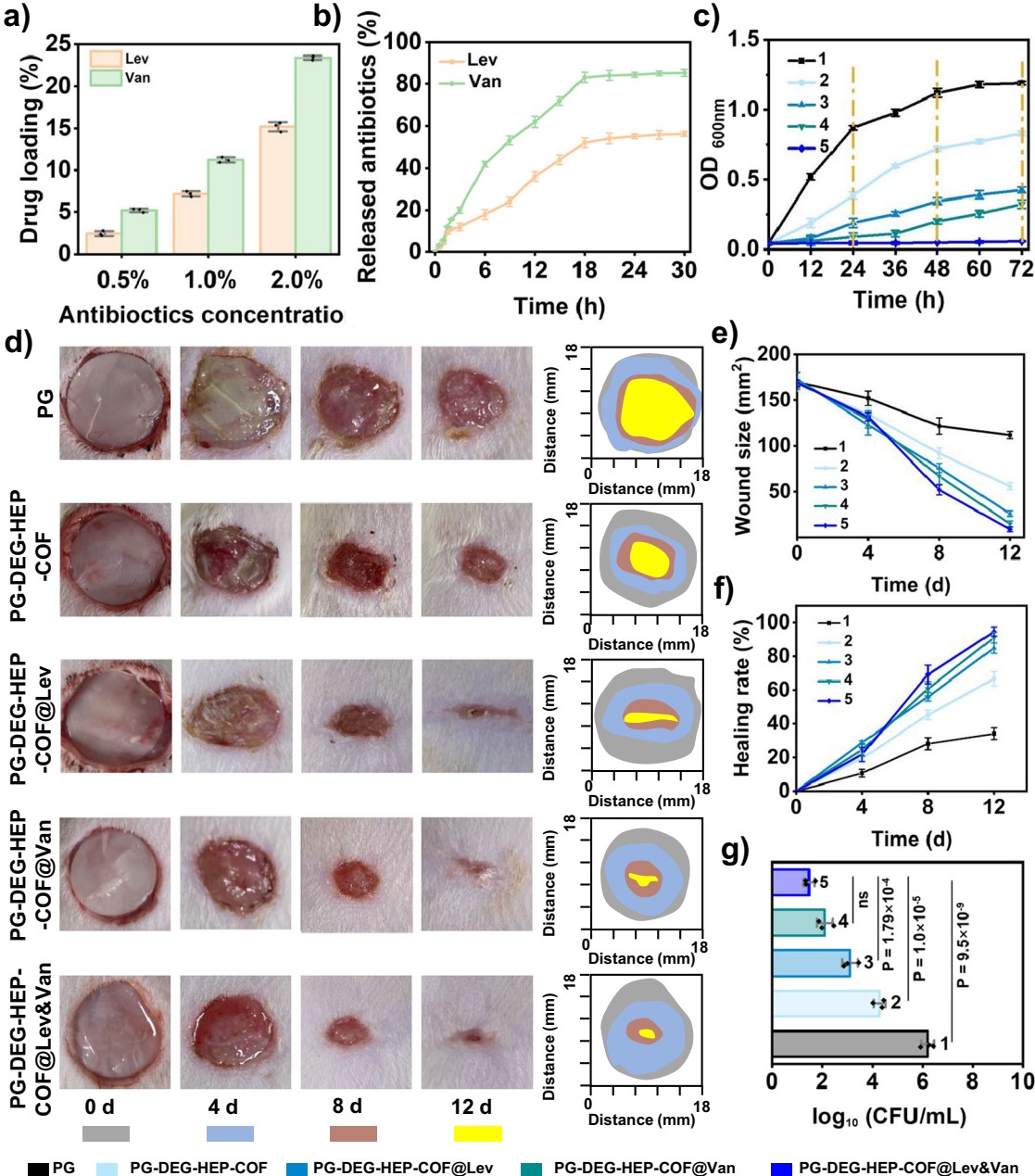

**Fig. 4 | The loading/releasing for antibiotics and the antibacterial effect in vitro and wound dressing in vivo. a** The antibiotic loading rate of DEG-HEP-COF. For each time point, $n = 3$ independent samples. **b** Antibiotic release efficiency of DEG-HEP-COF. For each group, $n = 3$ independent samples. **c** Evaluation of the antibacterial effect of electrospun membrane with/without DEG-HEP-COF@Antibiotics against MRSA by growth curve method for 12, 24, 36, 48, 60, and 72 h. For each time point, $n = 3$ samples. **d** Representative pictures (left) and traces (right) of wound healing for different groups at 0, 4, 8, and 12 d. For each group, three independent samples were recorded, and 1 representative image is shown. **e** Wound sizes for different groups during treatment. For each group, $n = 3$ independent samples. **f** Wound healing rates for different groups during treatment. For each group, $n = 3$ independent samples. **g** The colony counts of residual bacteria at the wounds from different groups on 12 d (The number of 1, 2, 3, 4, and 5 in **c**, **e**, **f**, and **g** represent the PG, PG-DEG-HEP-COF, PG-DEG-HEP-COF@Lev, PG-DEG-HEP-COF@Van, and PG-DEG-HEP-COF@Lev&Van membranes, respectively). For each group, $n = 3$ independent samples. The data are presented as mean ± SD. Statistical analysis was performed using one-way ANOVA analysis and ns means no significant difference. Source data are provided as a Source Data file.

ethylene glycol chains still exists in DEG-HEP-COF and can promote the crystallization process, and the presence of alkyl chains regulates the crystallization rate and pore pattern to improve the crystallinity. Additionally, the DEG+HEP-COF with random porous channels composed by alkoxy chains and alkyl chains does not possess the crystallinity, which further proves the structure of DEG-HEP-COF is different from the disordered structure of DEG+HEP-COF with random distribution of ethylene oxide chains and alkyl chains. The above-mentioned phenomena and calculation results indicate that the arrangement of the hetero-environmental pores in

DEG-HEP-COF is possibly alternant of hydrophobic alkyl chains and hydrophilic alkoxy chains.

Then, we further calculated the total energies of the tetragonal DEG-HEP-COF with different chain orientations and the different stacking structures by density-functional tight-binding method (DFTB+), including van der Waals dispersion, using Slater–Koster library in which O-N-H-C parameters were based on an early publication[46,47]. Regarding the intra-layer chain orientations of DEG-HEP-COF, the lowest relative total energy (RTE) of DEG-HEP-COF with all alkyl chains or alkoxy groups being arranged in the same pore reveals that it is the

most stable structure among all possible cases (Fig. 3e and Supplementary Fig. 38). Regarding the interlayer stacking structures, we calculated the relative crystal stacking energy (RSE) of DEG-HEP-COF with eclipsed and alternated AA stacking modes, which contain exactly the same and the reverse pore structures of the adjacent layers, respectively (Supplementary Fig. 39a, b). The calculation results show that the eclipsed AA stacking mode for DEG-HEP-COF is energy stabler than alternated AA stacking mode, indicating a significant energetic preference for constructing the 1D channels with only alkyl chains or alkoxy chains. Meanwhile, the energy of eclipsed AA stacking modes with slightly slipped structures was further calculated, and the results show that the RSE of slipped mode are unfavorable as compared with the eclipsed mode (Supplementary Fig. 39c, d). Therefore, the calculation indicates the rationality that the DEG-HEP-COF possesses the alternant pore structure.

We further designed and synthesized the dialdehyde monomers with a single side chain (hydrophobic alkyl chain or hydrophilic ethylene oxide chain) to construct the corresponding sCOFs (Supplementary Figs. 40–49). The pore size distribution data may infer the arrangement of side chains, which can further indirectly verify the structure of DEG-HEP-COF (Supplementary Figs. 50 and 51). Particularly, the pore size distribution analysis clearly exhibits two pore size distributions for the as-prepared sDEG-COFs (containing a single hydrophilic ethylene oxide chain in dialdehyde monomer), with 1.4 and 3.9 nm, which correspond to the small pores with the alkoxy chains, and large pores without the chains, respectively. The sHEP-COF (containing a single hydrophobic alkyl chain in dialdehyde monomer) exhibits a uniform pore size distribution with 1.7 nm. These phenomena prove that the alkoxy chains have the directional orientation of a stacking, which corresponds with the aforementioned arrangement inclination of the side chain. However, the COFs with a single side chain exhibit a low BET specific surface area and relatively weak crystallinity, which may result from the weak interaction of the single side chains with partial random orientations. Based on the above analysis, the proposed alternant pore structure may be reasonable and convincing.

## The application in co-delivery of hydrophilic and hydrophobic medicines

The combination of antibiotics is an important strategy to effectively combat drug-resistant bacterial infections. The wound healing process is often susceptible to infection by pathogenic bacteria such as *Escherichia coli*, *Staphylococcus aureus*, and drug-resistant bacteria, which can lead to the risk of persistent wound inflammation and sepsis. To achieve good bactericidal efficacy and maintain a long-term sterile environment of the wound, wound dressings are required to be able to release antibacterial drugs in an initial burst to rapidly kill bacteria at the wound site, followed by a long-term release to maintain an effective level of bacterial inhibition. The exudate from the wound facilitates the rapid release of the loaded hydrophilic drugs and the sustained release of the loaded hydrophobic drugs. However, loading one type of antibiotics cannot achieve a phased and controlled release, and the effect of antibacterial action is difficult to maintain. Therefore, the synergy of different antibiotics in the same delivery system can minimize antibiotic dosage and facilitate the process of dosage forms. The compatibility of different drugs with disparate properties (such as the different hydrophilicity) limits the combination of drugs[48,49]. The existing delivery systems loaded with hydrophilic and hydrophobic drugs have complex structures[50]. Considering the unique hetero-environmental porous structure of DEG-HEP-COF, we predicted it is a desired dual-antibiotic delivery vehicle. These synthesized COFs have no cytotoxicity to Huvecs and L929 cells by 3-(4,5-dimethylthiazol-2-yl)-2,5-diphenyltetrazolium bromide (MTT) assay (Supplementary Fig. 52.) Therefore, the COFs prepared in this work were used to load hydrophobic Lev and hydrophilic Van to explore the relationship between the pore environment and the loading/releasing behaviors of antibiotics.

The DEG-COF exhibited the high antibiotic loading amounts for hydrophilic Van (9.4%, 21.8% under 0.5% (w/v) and 2% (w/v) antibiotic conditions) and low loading amounts for hydrophobic Lev (1.3%, 2.1% under 0.5% (w/v) and 2% (w/v) antibiotic conditions). The HEP-COF shows the opposite trend (2.5%, 3.9% for Van and 5.1%, 21.6% for Lev under 0.5% (w/v) and 2% (w/v) antibiotic conditions, respectively) (Supplementary Fig. 53a, b, d, e). However, the DEG-HEP-COF exhibited high antibiotic loading amounts toward both Lev and Van. When the antibiotic usage increased from 0.5% (w/v) to 2% (w/v), the loading of Lev increased from 2.5% to 15.2% and that of Van increased from 5.2% to 23.4% (Fig. 4a and Supplementary Fig. 54). Significantly, the DEG+HEP-COF exhibited low antibiotic loading amounts toward both Lev and Van (Supplementary Fig. 53g, h). When the antibiotic usage increased from 0.5% (w/v) to 2% (w/v), the loading of Lev increased from 0.8% to 2.8% and that of Van increased from 1.2% to 3.2%. These results demonstrated that DEG-HEP-COF with the unique hetero-environmental porous structure could achieve high simultaneous loading of Lev and Van. The release rates of medicine are also important for the antimicrobial effects. The UV–Vis absorption spectra showed that the release rates of Lev and Van reached 56.2% and 85.2% after 30 h. These results demonstrated that DEG-HEP-COF could achieve a sustained release effect to maintain the long-term antibacterial activity of antibiotics (Fig. 4b). The release rates of medicine in DEG-COF and DEG + HEP-COF for Lev and Van reached about 80% after 30 h, which indicated that the sustained anti-infective effect is poor (Supplementary Table. 10 and Supplementary Fig. 53c, i). The release rates of medicine in HEP-COF for Lev and Van are different. However, the antibacterial effect is poor in an initial burst due to the low antibiotic loading amounts for hydrophilic Van (Supplementary Table 10 and Supplementary Fig. 53f). To sum up, the DEG-HEP-COF with hetero-environmental porous structure is ideal for the co-delivery of hydrophilic and hydrophobic medicines. Significantly, the results of loading and releasing for antibiotics of the COFs indirectly prove the rationality of the proposed hetero-environmental pores with the alternant arrangement.

The DEG-HEP-COF simultaneously loads both hydrophilic and hydrophobic antibiotics and achieves the sustained release effect to maintain the long-term antibacterial activity of antibiotics. So the DEG-HEP-COF carried antibiotic system was further studied on the antibacterial effect in vitro and wound dressing in vivo. Firstly, the cell viability and cytotoxicity of DEG-HEP-COF@Antibiotics were explored and the results indicate that DEG-HEP-COF@Antibiotics met the biological safety (Supplementary Figs. 55 and 56). Then, the polycaprolactone/gelatin (PG) membranes and PG membranes loaded with DEG-HEP-COF or DEG-HEP-COF@Antibiotics were made by electrospinning (named PG-DEG-HEP-COF, PG-DEG-HEP-COF@Lev, PG-DEG-HEP-COF@Van, and PG-DEG-HEP-COF@Lev&Van membranes) to investigate the antibacterial ability (Supplementary Fig. 57). As compared to membranes loaded with only Lev or Van, PG-DEG-HEP-COF@Lev&Van membranes showed the best antibacterial effect in vitro (Supplementary Fig. 58) and exhibited the excellent sustained antibacterial performances for 72 h (Fig. 4c). The full-thickness rat wound model of methicillin-resistant *S. aureus* (MRSA) infection was established to further evaluate the therapeutic effect of PG-DEG-HEP-COF@Antibiotics membranes as a potential wound dressing in vivo. The wound healing effect of the PG-DEG-HEP-COF@Lev&Van group was better than other treatment groups (Fig. 4d, e). After 12 d, the PG group showed a larger area of unhealed wounds than other groups. The wounds treated by PG-DEG-HEP-COF@Lev&Van membranes showed the best healing among these tested groups, which were covered by new epithelial tissue with a healing rate of 95% (Fig. 4f). The result was attributed to the synergistic antibacterial effect. The antibacterial effect of the different membranes was further assessed by counting the living MRSA at the wounds on 12 d. The PG-DEG-HEP-COF@Lev&Van group possessed the fewest colonies, indicating that inhibition of bacterial infection effectively promoted wound healing (Fig. 4g). Meanwhile,

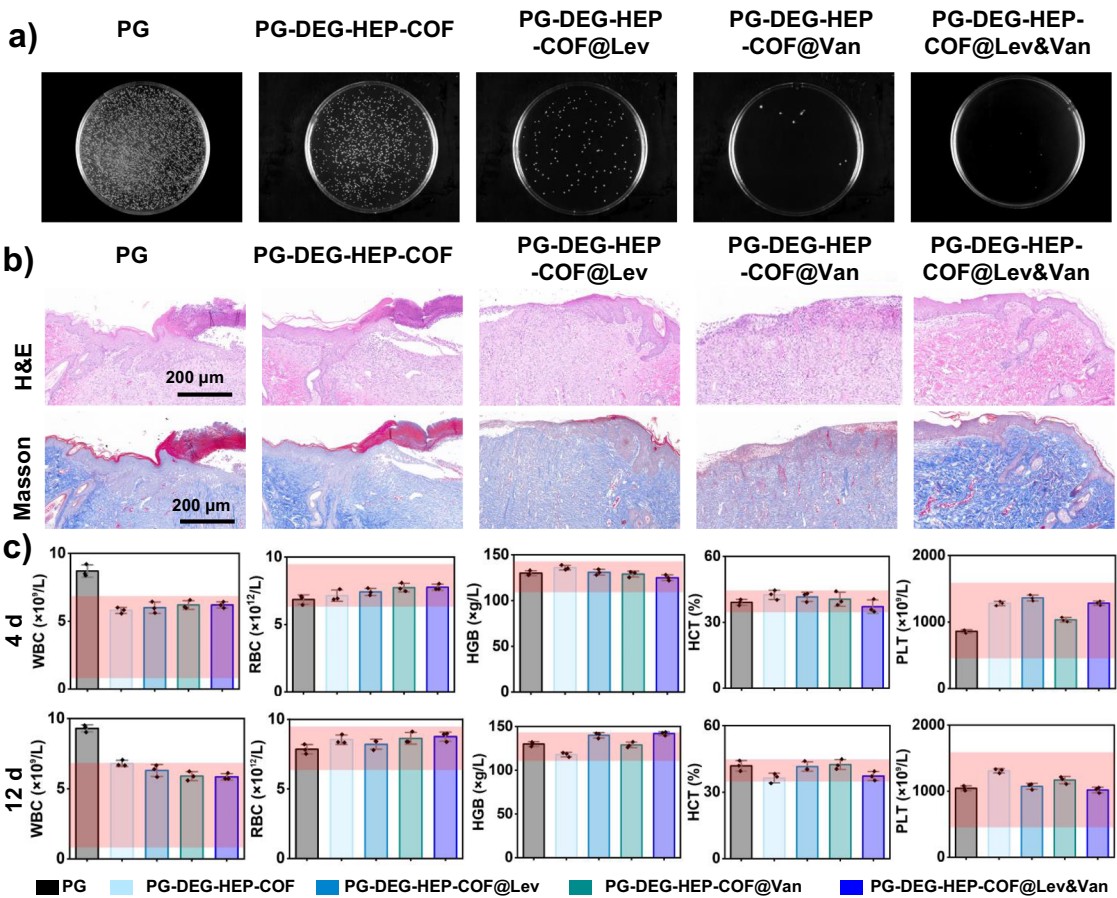

**Fig. 5 | Assessment of wound healing. a** Residual bacteria colonies on agar broth plates separated from wounds with different treatments at 12 d. **b** H&E and Masson staining sections of the wounds with different treatments at 12 d. Inflammatory cells (e.g. neutrophils and lymphocytes) were found in the PG group, while inflammation subsided in the PG-DEG-HEP-COF@Lev, PG-DEG-HEP-COF@Van, and PG-DEG-HEP-COF@Lev&Van groups, along with the presence of a large number of fibroblasts and elongated fibroblasts, which favored the formation of granulation tissue. Masson staining demonstrated collagen formation and distribution during wound healing. At 12 d, the wounds treated with PG-DEG-HEP-COF@Lev&Van membrane exhibits more collagen fibers (dark blue) than other groups, which are uniformly distributed and extended to the epidermis, indicating that the capability to facilitate wound healing of PG-DEG-HEP-COF@Lev&Van membrane is better than those of other test membranes. **c** Routine blood indicators at 4 d and 12 d (WBC, white blood cells; RBC, red blood cells; HGB, hemoglobin; HCT, red blood cell pressure product; PLT, platelets). For each group, $n = 3$ independent samples. At 4 and 12 d, the number of WBC in the PG-treated group exceeded the normal range, while all routine blood indicators were within the normal range for other membrane-treated groups. The shaded pink squares in Fig. 5c represent the normal range of routine blood indicators. For **a**, **b**, three independent samples were recorded, and one representative image is shown. The data in **c** are presented as mean ± SD. Source data are provided as a Source Data file.

routine blood, hematoxylin and eosin (H&E), and Masson trichrome staining results showed that the PG-DEG-HEP-COF@Lev&Van group reduced inflammation, promoted collagen deposition and epithelial formation to accelerate wound healing (Fig. 5). The H&E staining results demonstrated that the PG-DEG-HEP-COF@Lev&Van membrane-treated group obtained the maximum granulation tissue thickness (869 µm) at the 12 d among all these membranes treated groups (Supplementary Fig. 59a). In the meanwhile, the Masson staining results showed that the PG-DEG-HEP-COF@Lev&Van membrane-treated group exhibited the highest number of collagen fibers among all the groups (Supplementary Fig. 59b). These results indicate that the PG-DEG-HEP-COF@Lev&Van membrane possesses good ability to promote granulation regeneration, increase collagen content, and accelerate wound healing, therefore, shows high potential as a candidate for in vivo antimicrobial therapy and wound healing. Additionally, the antibacterial effect of COFs without loaded antibiotics was evaluated, which indicated that the primary effect on wound healing is attributed to the loaded antibiotics rather than the COFs themselves (Supplementary Fig. 60). In summary, the unique COF with the hetero-environmental pores provided an ideal environment for loading both hydrophilic and hydrophobic antibiotics, which enabled synergistic antibacterial effect and promoted wound healing. Furthermore, the unique structure of DEG-HEP-COF can be used in the fields of the combination of antitumor drugs, precision therapy, and personalized treatment.

## Discussion

The DEG-HEP-COF with hetero-environmental pores was designed and synthesized by the asymmetrical amphiphilic monomer with different side chains. The excellent porous and crystalline COF contains hydrophobic channels and hydrophilic channels. We systematically characterized and explored the chemical structure and growth process to investigate the differences between DGE-HEP-COF and other control COFs (DEG-COF, HEP-COF, and DEG+HEP-COF). In terms of the growth process, the experiment results indicate that the amphiphilic monomer possesses suitable crystallization rates. Meanwhile, theoretical calculation results suggest that the interaction of ethylene glycol chains exists in DEG-HEP-COF, which promotes the crystallization process, and the presence of alkyl chains regulates the crystallization rate and pore pattern to improve the crystallinity. The exploration of the growth process and the calculation indicate that the hydrophobic alkyl chain and hydrophilic alkoxy chain in the structure of DEG-HEP-COF are probably located in different pores. We further synthesized the

sDEG-COF and sHEP-COF with the single side chain (hydrophobic alkyl chain or hydrophilic ethylene oxide chain) to further indirectly verify the hetero-environmental pores structure from the pore size distribution. Furthermore, the DEG-HEP-COF exhibits low cytotoxicity and low hemolytic property, and can be utilized to load both hydrophobic and hydrophilic antibiotics for wound healing. This dual-antibiotic delivery strategy can expand the antibacterial range as compared with the single antibiotic ones, and delay or reduce the generation of drug resistance. In summary, the COF with hetero-environmental pores not only greatly expands the diversity of pore environments and manifests the significant influence of pore environments on the stacking mode and crystallinity but also provides a pathway for expanding the functionalities of COFs. The further exploration of COFs with multiple pore environments is in progress in our laboratory.

## Methods

### General synthesis procedure of COFs
In a typical procedure, the mixture of TAPP (27.0 mg, 0.04 mmol) and dialdehyde monomers (0.08 mmol), and mixture solvent with aqueous acetic acid (6 M) (ratio details are included in Supplementary Tables 2–5) was successively added into a 10 mL Pyrex tube. The mixture was then sonicated for 20 min. The Pyrex tube was degassed through three freeze–pump–thaw cycles. The tube was flame-sealed and heated at 120 °C for 72 h. The precipitate was collected by centrifugation, washed with anhydrous THF and acetone, and then dried under vacuum at 60 °C for 72 h to achieve the corresponding COF, including DEG-HEP-COF, DEG-COF, HEP-COF, and DEG+HEP-COF.

More experimental, analytical, and computational details are presented in the Supplementary Information.

### Statistical and reproducibility
Data were represented as means ± SD. All data were obtained from the results of at least three independent experiments. Results are from different samples rather than repeated measurements of the same sample. Statistical comparisons of all data were analyzed by Excel 2020 (Microsoft Office, USA), Origin 2018 (OriginLab, USA), ImageJ (National Institutes of Health, USA), or IBM SPSS Statistics 25 (International Business Machines Corporation, USA). $P$ values were determined by one-way ANOVA with Tukey's multiple comparisons post hoc test among multiple groups. Values of all significant correlations ($P < 0.05$) mean statistically significant and ns means no significant difference. For all studies, samples were randomly assigned to various experimental groups.

### Reporting summary
Further information on research design is available in the Nature Portfolio Reporting Summary linked to this article.

## Data availability
All data are available in the main text or the Supplementary Information. All data are available from the authors upon request. Source data are provided with this paper.

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

## Acknowledgements

B.-H.H. thanks for the financial support of the Strategic Priority Research Program of Chinese Academy of Sciences (Grant XDB36000000). X.D. acknowledges the financial support of the National Natural Science Foundation of China (Grants 22171057). B.-H.H. acknowledges the financial support of the National Natural Science Foundation of China (Grants 22161132010 and 22075060). We would like to appreciate Prof. Junliang Sun (Peking University) and Prof. Lang Jiang (Institute of Chemistry, Chinese Academy of Sciences) for their characterization efforts on electron diffraction and AFM, respectively.

## Author contributions

R.C., X.D., S.L., and B.-H.H. designed the project. W.J. prepared the COFs and measured the structure, composition, and performances of the materials. P.Z. performed the loading/releasing experiment for antibiotics and the antibacterial test in vitro and wound dressing in vivo. G.F. contributed to the STM test. Y.-Z.C. and T.-X.W. contributed to the SEM test and analysis. D.Y. focused on the simulation of the structure of COFs. W.J., Y.-Z.C., X.D., and B.-H.H. co-wrote the paper. All the authors discussed the results and commented on the manuscript.

## Competing interests

The authors declare no competing interests.
