## [Peer Review File · Nature Communications]

Covalent organic frameworks with hetero-environmental pores: Synthesis and co-delivery of hydrophilic and hydrophobic medicinesREVIEWER COMMENTS

Reviewer #1 (Remarks to the Author):

[Note from the editor - Please also see attached PDF]

This manuscript reported the synthesis of COFs with hetero-environmental pores through the one-step polymerization, and their performance investigation to load both hydrophobic and hydrophilic antibiotics for wound healing. The combination of antibiotics is a new and far-sighted strategy to effectively combat drug-resistant bacterial infections, which can minimize antibiotic dosage and facilitate the process of dosage forms. In my opinion, the concept of COF containing hetero-environmental pores is creative and very interesting to stimulate other researchers' enthusiasm for further exploration on its structural variety, exact characterization, and application. The work was well-designed and conducted, and the data as well as the manuscript were well-organized, therefore, I would like to recommend acceptance after some minor revisions. Questions to the researchers and some suggestions as follow:

1. Given the structural characteristics of the amphiphilic pores of DEG-HEP-COF, the simultaneous loading of different types of antibiotics for the combined treatment of wounds is an important application in this study. A more detailed description of the importance and advantages of antibiotic combination is needed in the introduction section based on corresponding literatures (e.g. Nat. Microbiol. 4, 1627-1635(2019); Science 367, 200-204(2020))
2. This manuscript explores the photo-degradation and photo-catalytic ability of materials based on free radical. Whether these radical have an impact on the wound healing?
3. As a potential functional framework, DEG-HEP-COF is modified with long side groups, which might be adverse to mass transfer in the channels of DEG-HEP-COF and further degrade the adsorption and photocatalysis performances. Please comment about that.
4. In Fig. 5c, DEG-HEP-COF with a heterogeneous environment was used as a carrier to evaluate the in vitro antibacterial effect, and only 24 h antibacterial results were showed. However, further observations for longer time such as 48 h or 72 h are more necessary to test the sustained antibacterial ability of the drug-loaded DEG-HEP-COF wound dressing.
5. In the treatment of wound healing, the thickness of granulation tissue and collagen fiber deposition can be more appropriately evaluated for wound recovery. In Supplementary Fig. 37, the authors only qualitatively demonstrated the distribution of collagen fibers by Masson staining, so it is recommended that the authors provide additional quantitative data on the thickness of granulation tissue and collagen fiber deposition for a more visual assessment of wound recovery.
6. In the section of Results Synthesis and characterizations of the chemical structure, the synthetic

method of DEG+HEP-COF lacks citation annotations like other products.

7. As shown in Fig. 3b, the scales of SEM images are not complete.

Reviewer #2 (Remarks to the Author):

The authors submitted "Synthesis of covalent organic frameworks with hetero-environmental pores and their application in co-delivery of hydrophilic and hydrophobic medicines" The authors did a lot of work in this manuscript and the results are very interesting. Here are my comments:

1. The authors selected a porphyrin unit to prepare COF materials. Why? this moiety is not new, so many COF are reported about this molecule.
2. The FTIR and high-resolution mass of all synthesized monomers in this study should be provided.
3. How about the stability of all COF materials in different acidic and basic mediums?
4. The reviewer did;t see XPS, SEM, TEM, and SEM-EDS data for all synthesized COF in this study.
5. How much drug loading inside COF materials?
6. what is the relationship between photocatalytic and drug delivery in this study? the authors should focus in one application and the manuscript's title should be changed.
7. FTIR figures in the SI should be provided with good shape and assign each peak.
8. How about the MTT assay data for all these synthesized COFs?
9. some references should be provided such as doi.org/10.1016/j.ccr.2023.215066; doi.org/10.1039/D0NJ04941G; doi.org/10.1039/D2NA00103A; doi.org/10.1039/D1MA00771H;

Reviewer #3 (Remarks to the Author):

The communication of Han et al. discusses the synthesis of a COF that has distinct hydrophobic and hydrophilic pockets. The COF is then characterized and applied as 1) photocatalyst for dye degradation, 2) photocatalyst for oxidative amine coupling and 3) vessel for drug delivery. Also the growth mechanism of the COFs is investigated.

So the paper contains a lot of information and applications and that is also the weakness of the work: all sections remain highly speculative and lack in depth characterization, resulting in a very lengthy paper, but relatively superficial.

Regarding the synthesis of the COFs (together with some control COFs), it remain unclear to me (in Fig 1) why this COF should have alternating pores with respectively the hydrophilic and -phobic chains alternatively pointing inwards. What is the proof for that, looking at the monomers the DEG-HEP-CHO can couple in both directions. As the COFs are stacked 2D COFs, they should have no rotational freedom

after forming, to orient the hydrophobic or hydrophilic chains.

Materials Studio optimizations are very effective for a quick in-house simulations, but more thorough calculations could be done there.

Authors claim high crystallinity of the COF, but Fig.2 does not show that: only one sharp reflection peak can be seen. I am also highly worried about the HR-TEM as it is known that HR-TEM electrons destroy the COFs and such measurements should be done using special techniques with very low dosages of electrons.

The growth mechanism in Figure 3 is to me completely speculative and the SEM pictures and XRD shown in Figure 3 certainly do not allow a discrimination of the two growth mechanisms.

In the photocatalytic experiments, obviously the porphyrine rings in the COF are big contributors to the overall activity. The authors report large differences in the excited state lifetimes based on fluorescence, but they do not provide an adequate reasoning. In the end, they conclude that it is due to the degree of crystallinity, so the variations and position of the hydrophobic/hydrophilic chains seem to have no effect, which renders the photocatalytic study a bit superfluous.

The oxidative coupling of amines (photocatalytically) is such an easy reaction that it almost always works.

Regarding the drug delivery, I don't really see why rat-wound experiments were necessary, but the conclusions that hydrophilic COFs will adsorb hydrophilic drugs and vice versa is a bit obvious.

Overall, the idea of the "checkboard" alternation of hydrophilic and hydrophobic pockets is a good idea. In the current paper, this is not proven but assumed or speculated. The high amount of application distracts from the core message.

Replies to Reviewers' Comments

Reviewer #1 (Remarks to the Author):

This manuscript reported the synthesis of COFs with hetero-environmental pores through the one-step polymerization, and their performance investigation to load both hydrophobic and hydrophilic antibiotics for wound healing. The combination of antibiotics is a new and far-sighted strategy to effectively combat drug-resistant bacterial infections, which can minimize antibiotic dosage and facilitate the process of dosage forms. In my opinion, the concept of COF containing hetero-environmental pores is creative and very interesting to stimulate other researchers' enthusiasm for further exploration on its structural variety, exact characterization, and application. The work was well-designed and conducted, and the data as well as the manuscript were well-organized, therefore, I would like to recommend acceptance after some minor revisions. Questions to the researchers and some suggestions as follow:

Thank Reviewer 1 very much for his/her insightful review and recommendation of publication after minor revision.

1. *Given the structural characteristics of the amphiphilic pores of DEG-HEP-COF, the simultaneous loading of different types of antibiotics for the combined treatment of wounds is an important application in this study. A more detailed description of the importance and advantages of antibiotic combination is needed in the introduction section based on corresponding literatures (e.g. Nat. Microbiol. 4, 1627-1635(2019); Science 367, 200-204(2020))*

In response to his/her comment, we have thoroughly reviewed the corresponding literature and incorporated these findings into the Introduction section. The relevant literatures have been cited as Refs. 31–32 in the revised manuscript. The details are shown as follows.

“Antibiotic-resistant bacteria pose a significant threat to human health. The development of new antibiotics lags behind the evolution of bacterial resistance, highlighting the urgent need to effectively utilize existing antibiotics. Aminoglycoside/beta-lactam combinations exhibit synergistic effects and are the most commonly used combinations in clinical practice (Nat. Microbiol. 2019, 4, 1627–1635). By combining antibiotics with different mechanisms of action, we can overcome resistance mechanisms of bacteria and increase the effectiveness of treatment (Science 2020, 367, 200–204).”

2. *This manuscript explores the photo-degradation and photo-catalytic ability of materials based on free radical. Whether these radical have an impact on the wound healing?*

According to his/her suggestions, we have performed some corresponding experiments. The results indicate that DEG-HEP-COF, DEG-COF, HEP-COF, and DEG+HEP-COF do not exhibit significant bactericidal effects against MRSA under dark/light conditions. These results suggest that the reactive oxygen species generated by the materials are insufficient to kill the bacteria, and therefore, they do not contribute positively to the treatment of bacterial infections in wound healing. The primary effect on wound healing is attributed to the loaded antibiotics. The corresponding details have been added to the revised manuscript and Supplementary Information.

Supplementary Fig. 38. (a) Evaluation of the antibacterial effect of COFs ($100 \mu\text{g mL}^{-1}$) against MRSA by turbidity method. (b) Evaluation of the antibacterial effect of COFs against MRSA by disc diffusion method. The diameter of the inhibition zone can be utilized to evaluate the antibacterial activity. (The numbers 1, 2, 3, 4, and 5 represent the blank, DEG-HEP-COF, DEG-COF, HEP-COF, and DEG+HEP-COF, respectively. The a, b, c, d, and e represent the COFs concentration of 50, 100, 200, 400, $1000 \mu\text{g}\cdot\text{mL}^{-1}$, respectively).

3. As a potential functional framework, DEG-HEP-COF is modified with long side groups, which might be adverse to mass transfer in the channels of DEG-HEP-COF and further degrade the adsorption and photocatalysis performances. Please comment about that.

Thank Reviewer 1 very much for the suggestion. In this manuscript, related problems were considered in the design process, so the dialdehyde monomer was designed as a terphenyl rather than the frequently used terephthalaldehyde to avoid the complete pore blockade. The structure of COF with side chains of different lengths has also been simulated, and the result shows that the long chain used in this work does not block the pores. The obtained DEG-HEP-COF with side chains also shows high BET specific surface area up to $1030 \text{ m}^2 \text{ g}^{-1}$, and the pore size is centered at 1.4 nm. Meanwhile, a series of fluorinated oligoamide nanorings with internal diameters ranging from 0.9 to 1.9 nm have been reported in the previous literature (*Science* **2022**, *76*, 698–699.), showing that individual water molecules flow through the small channels faster than through the large ones. At the same time, Jiang group also reported the effect of a series of COFs with different pore sizes on water mass transfer (*Nat.*

Commun. **2021**, *12*, 6747 (1–10)). The results showed that microporous COFs with small pores could deliver quickly water at low pressures by synergistic nucleation and capillary condensation. Therefore, the presence of side chains does not degrade the adsorption and photocatalysis performances.

4. In Fig. 5c, DEG-HEP-COF with a heterogeneous environment was used as a carrier to evaluate the *in vitro* antibacterial effect, and only 24 h antibacterial results were showed. However, further observations for longer time such as 48 h or 72 h are more necessary to test the sustained antibacterial ability of the drug-loaded DEG-HEP-COF wound dressing.

According to his/her suggestion, we have conducted additional antibacterial experiments on the drug-loaded DEG-HEP-COF wound dressing for 48 and 72 h *in vitro*. The results demonstrate that the drug-loaded DEG-HEP-COF wound dressing exhibits excellent sustained antibacterial properties for 72 h. The corresponding details have been added to the revised manuscript.

Fig 4c. Evaluation of the antibacterial effect of the electrospun membrane with/without DEG-HEP-COF@Antibiotics against MRSA by growth curve method for 12, 24, 36, 48, 60, and 72 h. (The number of 1, 2, 3, 4, and 5 represent the PG, PG-DEG-HEP-COF, PG-DEG-HEP-COF@Lev, PG-DEG-HEP-COF@Van, and PG-DEG-HEP-COF@Lev&Van membranes, respectively).

5. In the treatment of wound healing, the thickness of granulation tissue and collagen fiber deposition can be more appropriately evaluated for wound recovery. In Supplementary Fig. 37, the authors only qualitatively demonstrated the distribution of collagen fibers by Masson staining, so it is recommended that the authors provide additional quantitative data on the thickness of granulation tissue and collagen fiber deposition for a more visual assessment of wound recovery.

Per suggestions, we have added quantitative data on granulation tissue thickness and collagen fiber deposition to demonstrate the wound healing process more visually. The corresponding details have been added to the revised manuscript and Supplementary Information.

“The H&E staining results demonstrated that the PG-DEG-HEP-COF@Lev&Van membrane treated group obtained the maximum granulation tissue thickness (869 μm) on the 12th day among all these membranes treated groups (Supplementary Fig. 37a). In the meanwhile, the Masson staining results showed that the PG-DEG-HEP-COF@Lev&Van membrane treated group exhibited the highest number of collagen fibers among all the groups (Supplementary Fig. 37b). These results indicate that the PG-DEG-HEP-COF@Lev&Van membrane possesses good ability to promote granulation regeneration, increase collagen content, and accelerate wound healing, therefore, showing high potential as a candidate for in vivo antimicrobial therapy and wound healing.”

Supplementary Fig. 37. Quantitative analysis of H&E and Masson staining sections acquired from different groups to calculate the granulation tissue thickness and collagen deposition ($n = 3$) (The number of 1, 2, 3, 4, and 5 represent the PG, PG-DEG-HEP-COF, PG-DEG-HEP-COF@Lev, PG-DEG-HEP-COF@Van, and PG-DEG-HEP-COF@Lev&Van membranes, respectively).

6. In the section of Results Synthesis and characterizations of the chemical structure, the synthetic method of DEG+HEP-COF lacks citation annotations like other products.

According to his/her suggestion, the detailed synthetic method has been added to the revised manuscript.

“DEG+HEP-COF with randomly distributed alkyl and alkoxy chains via copolymerization of TAPP, DEG-CHO, and HEP-CHO (Fig.1) (Detailed synthesis method are described in Supplementary Schemes 3 and 4).”

7. As shown in Fig. 3b, the scales of SEM images are not complete.

Per suggestion, we have completed the scales of SEM images in the revised manuscript. According to the logic of the manuscript, the Fig. 3b was renumbered as Supplementary Fig. 18.

Supplementary Fig. 18. SEM images of (a) DEG-HEP-COF and (b) DEG-COF recorded at different reaction time.

Reviewer #2 (Remarks to the Author):

The authors submitted "Synthesis of covalent organic frameworks with hetero-environmental pores and their application in co-delivery of hydrophilic and hydrophobic medicines" The authors did a lot of work in this manuscript and the results are very interesting. Here are my comments:

Thank Reviewer 2 very much for his/her insightful comments. According to his/her suggestions, we have performed some corresponding experiments to further explore the structures and properties of these materials and revised our manuscript carefully. We believe this manuscript has been significantly improved, and hope the Reviewer satisfies our revision effort.

1. The authors selected a porphyrin unit to prepare COF materials. Why? this moiety is not new, so many COF are reported about this molecule.

Thank Reviewer 2 very much for the suggestion. Indeed, the porphyrin unit has been frequently utilized to synthesize COFs. However, the novelty and emphasis of the manuscript lies in the hetero-environmental pores of COF by designing and synthesizing the asymmetrical amphiphilic monomer with different side chains rather than porphyrin unit. Here we choose porphyrin as the building unit only because its C_4 symmetry ensures the possibility to construct the rational checkered structure. Meanwhile, at our best knowledge, those units with C_3 and C_6 symmetry cannot produce the checkered pore structure.

2. The FTIR and high-resolution mass of all synthesized monomers in this study should be provided.

According to his/her suggestion, the FT-IR and high-resolution mass spectra of all synthesized monomers in this study have been added.

Supplementary Fig. 1. FT-IR spectra of synthesized monomers in this study.

Supplementary Table 1. The assignment of wavenumber peaks in FT-IR spectra (Supplementary Fig. 1).

Wavenumber (cm ⁻¹)	DEG-HEP-CHO	DEG-CHO	HEP-CHO	sDEF-CHO	sHEP-CHO
$\nu_s(-CH_3)$, $\nu_{as}(-CH_3)$	2930, 2813	2825	2925, 2795	2881	2922, 2805
$\nu_s(-CH_2)$, $\nu_{as}(-CH_2)$	2860, 2725	2875, 2739	2861, 2710	2885, 2735	2853, 2740
$\nu(-C=O)$	1701	1683	1696	1689	1692
$\nu(-C=C)$	1602	1597	1594	1594	1602
$\nu(C-O-C)$	1105	1102	—	1096	—

HR-MS spectra of monomers.

3. How about the stability of all COF materials in different acidic and basic mediums?

The COFs based on Schiff bases are unstable in aqueous HCl (1 M) and NaOH (1 M) solutions owing to the disassociation of imine linkages.

Supplementary Fig. 12. PXRD patterns of all COFs materials before (a), and after treatments in aqueous 1 M HCl (b) and 1 M NaOH (c) solutions for 1 day.

4. The reviewer didn't see XPS, SEM, TEM, and SEM-EDS data for all synthesized COF in this study.

Per suggestion, the XPS, SEM, TEM, and SEM-EDS data have been provided in the revised manuscript and Supplementary Information.

Supplementary Fig. 8. XPS investigation of synthesized COFs in this study. High-resolution XPS spectra of C 1s (a), N 1s (b), and O 1s (c) for COFs.

Supplementary Fig. 17. SEM and SEM-EDS images of DEG-HEP-COF (a), DEG-COF (b), HEP-COF (c), and DEG+HEP-COF (d).

Supplementary Fig. 27. SEM and SEM-EDS images of sDEG-COF (a) and sHEP-COF (b).

Supplementary Fig. 15. HR-TEM images of DEG-HEP-COF (a), DEG-COF (b), HEP-COF (c), and DEG+HEP-COF (d).

5. How much drug loading inside COF materials?

According to his/her suggestion, we have further explored the loading and releasing capacity of DEG-COF, HEP-COF, and DEG+HEP-COF for different drugs. To sum up, the DEG-HEP-COF with hetero-environmental pore structure is ideal for the co-delivery of both hydrophilic and hydrophobic medicines. The data have been added to the revised Supplementary Information as follows.

Supplementary Table 10. Drug loading of COF materials in different antibiotic concentrations

Antibiotic concentration (g mL ⁻¹)	Antibiotic	Drug loading (μg mg ⁻¹)			
		DEG-COF	HEP-COF	DEG-HEP-COF	DEG+HEP-COF
0.5%	Lev	12.1	52.9	25.6	32.0
	Van	101.3	25.6	54.9	43.8
1.0%	Lev	20.4	124.3	77.6	46.0
	Van	171.1	33.1	126.1	54.9
2.0%	Lev	21.5	275.5	179.2	48.2
	Van	278.8	40.1	314.1	59.3

6. what is the relationship between photocatalytic and drug delivery in this study? the authors should focus in one application and the manuscript's title should be changed.

Thank Reviewer 2 very much for his/her insightful comments. After reappraising the significance of the photocatalysis application part, we fully agree with his/her suggestions. The simple photocatalytic research described as an example in the

previous manuscript preliminarily proved the advantages of the DEG-HEP-COF. However, the emphasis of this manuscript is to demonstrate the advantages of the hetero-environmental pores. To enhance the clarity of the article's logic, we have reconstructed the manuscript, deleted the photocatalytic part and focused on drug delivery based on the unique pore structure of DEG-HEP-COF by specifically delving into the investigation of the relationship between the pore environment and the loading and releasing of antibiotics to improve the logic and highlight the key points in the revised manuscript. Furthermore, the impact of pore environment on the optical and catalytic properties of materials will be extensively explored in the forthcoming research.

7. FTIR figures in the SI should be provided with good shape and assign each peak.

According to his/her suggestion, we have retested the FT-IR spectra of COFs, and the 2000–1000 cm^{-1} fragment is magnified. The key peaks have been assigned in the spectra, and the results have been added to the Supplementary Information as follows.

Supplementary Fig. 6. FT-IR spectra of DEG-HEP-COF, DEG-COF, HEP-COF, and DEG+HEP-COF.

8. How about the MTT assay data for all these synthesized COFs?

Per suggestion, we investigated the cytotoxicities of these synthesized COFs on Huvecs and L929 cells. The results showed that when co-culturing the cells with COFs for 24 hours, there were no significant differences in cell viability between DEG-COF, HEP-COF, DEG-HEP-COF, DEG+HEP-COF, and the control group. MTT assay indicated these synthesized COFs have no or little cytotoxicity to Huvecs and L929 cells (Supplementary Fig. 30). The details are as follows.

Supplementary Fig. 30. Evaluation of the cytotoxicity of these synthesized COFs on HUVECs (a) and L929 cells (b) by CCK method.

9. some references should be provided such as doi.org/10.1016/j.ccr.2023.215066;
doi.org/10.1039/D0NJ04941G;
doi.org/10.1039/D1MA00771H;
doi.org/10.1039/D2NA00103A;

Thank Reviewer 2 very much for the suggestion. We carefully read and summarized the relevant literature you recommended, which are cited as Refs. 9, 13, 29, and 30 in the revised manuscript.

Reviewer #3 (Remarks to the Author):

The communication of Han et al. discusses the synthesis of a COF that has distinct hydrophobic and hydrophilic pockets. The COF is then characterized and applied as 1) photocatalyst for dye degradation, 2) photocatalyst for oxidative amine coupling and 3) vessel for drug delivery. Also the growth mechanism of the COFs is investigated. So the paper contains a lot of information and applications and that is also the weakness of the work: all sections remain highly speculative and lack in depth characterization, resulting in a very lengthy paper, but relatively superficial.

Thank Reviewer 3 very much for his/her insightful comments. After reappraising the significance of the photocatalysis application part, we fully agree with his/her suggestions. The simple photocatalytic research described as an example in the previous manuscript preliminarily proved the advantages of the DEG-HEP-COF. However, the emphasis of this manuscript is to demonstrate the advantages of the hetero-environmental pores. To enhance the clarity of the article's logic, we have reconstructed the manuscript, deleted the photocatalytic part and focused on drug delivery based on the unique pore structure of DEG-HEP-COF by specifically delving into the investigation of the relationship between the pore environment and the loading and releasing of antibiotics to improve the logic and highlight the key points in the revised manuscript. Furthermore, the impact of pore environment on the optical and catalytic properties of materials will be extensively explored in the forthcoming research. Thank Reviewer 3, I hope the revised manuscript gets much stronger.

1. Regarding the synthesis of the COFs (together with some control COFs), it remains unclear to me (in Fig 1) why this COF should have alternating pores with respectively the hydrophilic and -phobic chains alternatively pointing inwards. What is the proof for that, looking at the monomers the DEG-HEP-CHO can couple in both directions. As the COFs are stacked 2D COFs, they should have no rotational freedom after forming, to orient the hydrophobic or hydrophilic chains.

Thank Reviewer 3 very much for his/her insightful comments. The COFs are synthesized based on dynamic covalent chemistry (Schiff base reaction in this work). One of the key advantages of dynamic covalent chemistry is the reversibility. Dynamic covalent chemistry in COFs can enable self-correction properties, where defected COF structures can undergo reversible reactions to repair themselves. Dynamic covalent chemistry results in the generated COFs with the desired lowest energy structure in thermodynamic equilibrium. Furthermore, the ability of COFs to undergo self-correction and rotational motion can be influenced by various factors, including the intermolecular interactions between the layers, the flexibility of the organic linkers, and the presence of guest molecules or solvents within the COF structures. (Such as COF-to-COF transformation to prepare microtubular COFs and neutral COFs conveniently transformation to prepare ionic COFs indicated the self-correction and linker transformation occurs. *Angew. Chem. Int. Ed.*, **2023**, 63, e202300373; *Angew. Chem. Int. Ed.*, **2022**, 61, e202117390.) Therefore, the optimized synthesis conditions and suitable monomers are necessary for preparing highly crystalline COF materials. Meanwhile, the alkoxy chain is conducive to crystallization, which is analogous to the good crystallization behavior of polyethylene glycol that has been studied in previous researches. On the basis of chemistry principles and dynamic covalent chemistry, the hydrophobic and hydrophilic chains will probably locate in the right pores with the lowest energy. (hydrophilic chains like to get together and hydrophobic chains, too. This is the chemical principle of micelle formation.)

This manuscript from the perspective of crystallinity analyses, energy calculation, and double antibiotic loading application research of COFs proves the structure rationality of the proposed hetero-environmental pores. The obvious differences in the growth processes demonstrated the regulation of the branch chains on the structure and performance of COFs. Meanwhile, the STM with the atomic resolution was conducted to tentatively study the arrangement of the two side chain groups. From STM results, we can determine the location of TAPP and the structure of surface DEG-HEP-COF, but still cannot directly observe the side chain arrangement of DEG-HEP-COF. Hence the theoretical calculations were carried out to further explore the structure of DEG-HEP-COF. The calculation indicates the rationality of the alternant pore structure in the DEG-HEP-COF. We further synthesized the dialdehyde monomers with a single side chain (hydrophobic alkyl chain or hydrophilic ethylene oxide chain) to construct the sCOFs (Supplementary Schemes 5 and 6, Supplementary Figs. 26 and 27). The pore size distribution data infer the alkoxy chains have the directional orientation of a stacking. Significantly, the results of co-loading and releasing antibiotics of the DEG-HEP-COF indirectly prove the rationality of the proposed

hetero-environmental pores.

2. Materials Studio optimizations are very effective for a quick in-house simulations, but more thorough calculations could be done there.

Thank Reviewer 3 very much for his/her feedback on Materials Studio's optimization capabilities. Materials Studio's optimization algorithms are effective for quick in-house simulations, providing rapid insights into materials' properties. However, more thorough calculations requiring higher accuracy, the software requires the integration of external quantum chemistry software to perform advanced quantum mechanical calculations. In this manuscript, we firstly calculated the interlayer interaction energy of the DEG/HEP chains of COFs' fragments. Then, we further calculated the total energies of the tetragonal DEG-HEP-COF with different chain orientations and the different stacking structures by density-functional tight-binding method (DFTB+), including van der Waals dispersion, using Slater–Koster library in which O-N-H-C parameters were based on an early publication. The calculation indicates the rationality of the alternant pore structure in DEG-HEP-COF possesses. The details are shown as follows.

“We firstly calculated the interlayer interaction energy of the DEG/HEP chains of COFs' fragments. The fragment of DEG-COF presents the most favorable interaction energy among these three samples (Supplementary Fig. 23), which is beneficial to the stacking structure of COFs to promise the crystallinity. It can be speculated that the alkoxy chain is conducive to crystallize, which may be analogy to the good crystallization behavior of polyethylene glycol that has been studied in previous researches. This result shows good agreement with the phenomenon in above-mentioned formation process study that the DEG-COF containing hydrophilic ethylene oxide chains exhibits a fast crystallization and high crystallinity. In contrast, the fragment of HEP-COF presents the weakest interaction energy among these three samples, and the low crystallinity of HEP-COF probably originates from that the alkyl chains severely affect the stacking and crystallizing process. From the structural perspective, the DEG-HEP-COF containing half of the alkyl chain may exhibit weakened crystallinity to a certain extent. However, the DEG-HEP-COF exhibits the outstanding crystallinity and even better than that of DEG-COF. Herein, the interaction of ethylene glycol chains still exists in DEG-HEP-COF and can promote the crystallization process, and the presence of alkyl chains regulates the crystallization rate and pore pattern to improve the crystallinity. Additionally, the DEG+HEP-COF with random porous channels composed by alkoxy chains and alkyl chains does not possess the crystallinity, which further proves the structure of DEG-HEP-COF is different from the disordered structure of DEG+HEP-COF with random distribution of ethylene oxide chains and alkyl chains. The above-mentioned phenomena and calculation indicate that the arrangement of the hetero-environmental pores in DEG-HEP-COF is possibly alternant of hydrophobic alkyl chain and hydrophilic alkoxy chain.

Then, we further calculated the total energies of the tetragonal DEG-HEP-COF with different chain orientations and the different stacking structures by density-functional

tight-binding method (DFTB+), including van der Waals dispersion, using Slater–Koster library in which O-N-H-C parameters were based on an early publication. Regarding the intra-layer chain orientations of DEG-HEP-COF, the lowest relative total energy (RTE) of DEG-HEP-COF with all alkyl chains or alkoxy groups being arranged in the same pore reveals that it is the most stable structure among all possible cases (Fig. 3d and Supplementary Fig. 24). Regarding the inter-layer stacking structures, we calculated the relative crystal stacking energy (RSE) of DEG-HEP-COF with eclipsed and alternated AA stacking modes, which contain exactly the same and the reverse pore structures of the adjacent layers, respectively (Supplementary Fig. 25a and b). The calculation results show that the eclipsed AA stacking mode for DEG-HEP-COF is energy stabler than alternated AA stacking mode, indicating a significant energetic preference for constructing the 1D channels with only alkyl chains or alkoxy chains. Meanwhile, the energy of eclipsed AA stacking modes with slightly slipped structures was further calculated, and the results show that the RSE of slipped mode are unfavorable as compared with the total eclipsed mode (Supplementary Fig. 25c and d). Therefore, the calculation indicates the rationality that the DEG-HEP-COF possesses the alternant pore structure.”

3. Authors claim high crystallinity of the COF, but Fig.2 does not show that: only one sharp reflection peak can be seen. I am also highly worried about the HR-TEM as it is known that HR-TEM electrons destroy the COFs and such measurements should be done using special techniques with very low dosages of electrons.

Thank Reviewer 3 very much for his/her insightful comments. The sharp and intense diffraction peak is typically observed in well-crystallized COFs, indicating a highly ordered structure with long-range periodicity. By analyzing the position and shape of the diffraction peak, researchers can determine the lattice parameters, symmetry, and crystal structure of the COFs. The PXRD pattern of a crystalline material typically consists of multiple diffraction peaks corresponding to the arrangement of atoms within the crystal lattice. However, it is common to observe only one prominent peak in COFs' PXRD patterns. In the case of COFs, the organic molecules within the framework are relatively large and typically have a complex structure. As a result, the number of distinct crystallographic planes that can diffract X-rays and produce observable peaks is limited. Furthermore, the COFs are generally isolated as powders comprising aggregated nanometer-scale crystallites rather than the single-crystalline, micrometer-sized particles. The presence of imperfections in COF powders can further contribute to the broadening and attenuation of diffraction peaks, potentially reducing the visibility of additional peaks that might otherwise be present. (In fact, so far, there are only very limited COFs can be obtained single-crystals for precise characterization. *Science*, **2018**, 361, 48–52. *Nature Chemistry*, **2023**, 15, 841–847.) In fact, the DEG-HEP-COF exhibits excellent crystallinity with a strong and sharp diffraction peak.

HR-TEM is indeed a powerful technique, in which the energetic electrons may decompose the COFs. To mitigate this issue, special techniques can be used to minimize the damage to COFs during HR-TEM measurements including low-dose

imaging or low-energy imaging. The transmission electron microscope FEI Talos F200S (200 kV) was used to measure the COFs in this work.

4. The growth mechanism in Figure 3 is to me completely speculative and the SEM pictures and XRD shown in Figure 3 certainly do not allow a discrimination of the two growth mechanisms.

Thank Reviewer 3 very much, we have carefully considered his/her concerns regarding the growth mechanism presented in Figure 3. With respect to the SEM pictures and XRD data shown in Figure 3, our intention was to provide a preliminary characterization of the samples, to illustrate the differences of morphological and crystallinity features observed during the growth process, and to match a plausible mechanism based on the available literatures and preliminary observations. However, we understand that further experimental evidence and analysis are necessary to validate and support this mechanism. In light of his/her comment, we deleted the mechanism models and highlight the differences of growth process between DEF-HEP-COF and DEG-COF to improve the clarity and scientific rigour of manuscript. The obvious differences in the growth processes demonstrated the regulation of the branch chains on the structure and performance of COFs.

Fig. 3. Growth process and the pore structure verification of COFs. (a) BET specific surface area values and PXRD patterns of DEG-HEP-COF recorded at different reaction time. (b) BET specific surface area values and PXRD patterns of DEG-COF recorded at different reaction time. (c) SEM images of DEG-HEP-COF recorded at 14 d. (d) STM image of DEG-HEP-COF. STM images obtained by co-condensation of TAPP and DEG-HEP-CHO with thermal annealing in water atmosphere. Imaging parameters: $I_{set} = 60$ pA, $V_{bias} = -0.6$ V. (e) The RTE of tetragons with different chain orientations. Case-4 shows four alkyl chains in the same hole. Case-3 shows three alkyl chains in the same hole. Case-2-1 and Case-2-2 shows two alkyl chains in the same hole, which are in ortho-position and in para-position, respectively. (The high-resolution version of Fig. 3e is included in the Supplementary information, Supplementary Fig. 24).

5. In the photocatalytic experiments, obviously the porphyrine rings in the COF are big contributors to the overall activity. The authors report large differences in the excited state lifetimes based on fluorescence, but they do not provide an adequate reasoning. In the end, they conclude that it is due to the degree of crystallinity, so the variations and position of the hydrophobic/hydrophilic chains seem to have no effect, which renders the photocatalytic study a bit superfluous.

Thank Reviewer 3 very much for his/her insightful comments. After carefully considering his/her suggestions, to enhance the clarity of the article's logic, we have reconstructed the manuscript, deleted the photocatalytic part and focused on drug delivery based on the unique pore structure of DEG-HEP-COF by specifically delving into the investigation of the relationship between the pore environment and the loading and releasing of antibiotics to improve the logic and highlight the key points in the revised manuscript.

However, the separation and transfer of photogenerated charges are the key processes of this photocatalytic reaction. The designed amphiphilic monomer in this manuscript is an asymmetric monomer, and the presence of hydrophobic and hydrophilic chains affects the polarity of the molecule. According to previous researches, polarity is an attractive character in organic semiconducting molecules, which intrinsically affects their light harvesting, spin-orbital interaction, and photo-induced charge generation and separation (*J. Am. Chem. Soc.* **2014**, *136*, 17802–17807; *Angew. Chem. Int. Ed.* **2018**, *57*, 14188–14192; *J. Am. Chem. Soc.* **2011**, *133*, 13437–13444; *Inorg. Chem.* **2020**, *59*, 3142–3151.). The high crystallinity of DEG-HEP-COF is beneficial to efficiently separate and transfer the photogenerated charges, which can reduce the number of carrier recombination and endow DEG-HEP-COF with efficiently photocatalytic performance. Indeed, there is a correlation between structure, crystallinity, and properties. However, the current conclusions lack precise substantiation.

6. The oxidative coupling of amines (photocatalytically) is such an easy reaction that it almost always works.

Thank Reviewer 3 very much for his/her insightful comments. The degradation capability of dyes indicated the DEG-HEP-COF possesses excellent catalytic activity in water systems. To take full advantage of the photophysical property, electrochemical behavior, and the amphiphilic porous structure of DEG-HEP-COF, we explore its photocatalytic activity in the organic system by choosing benzylamine oxidation reaction as an example. After reappraising the significance of the photocatalysis application part, we fully agree with his/her suggestions that the emphasis of this manuscript is to demonstrate the advantages of the hetero-environmental pores. To enhance the clarity of the article's logic, we have reconstructed the manuscript, deleted the photocatalytic part and focused on drug delivery based on the unique pore structure of DEG-HEP-COF by specifically delving into the investigation of the relationship between the pore environment and the loading and releasing of antibiotics to improve the logic and highlight the key points in the revised manuscript. Furthermore, the impact of pore environment on the optical and catalytic properties of materials will be extensively explored in the forthcoming research.

7. Regarding the drug delivery, I don't really see why rat-wound experiments were necessary, but the conclusions that hydrophilic COFs will adsorb hydrophilic drugs and vice versa is a bit obvious.

Thank Reviewer 3 very much for his/her insightful comments, and we understand his/her concern regarding the rat-wound experiments and the apparent obviousness of the conclusions. Combining antibiotics with different/opposite properties can overcome the resistance mechanisms of bacteria and increase the effectiveness of treatment. However, the compatibility of different antibiotics limits the combination of drugs. The focus of this manuscript is to load dual antibiotics with disparate properties (such as the different hydrophilicity) based on DEG-HEP-COF unique hetero-environmental pores. In drug delivery, it is important to consider not only the adsorption capacity of the drug, but also the release behavior of the drug. Therefore, relevant experimental validation is necessary to ensure the efficacy and feasibility of using COFs as drug delivery systems. The result is difficult to predict directly. For example, from a structural perspective, both DEG-HEP-COF and DEG+HEP-COF contain hydrophilic and hydrophobic regions. However, experimental results show obvious differences between the two COFs, indirectly demonstrating the role of structural regularity in the absorption and release of hydrophilic and hydrophobic drugs.

The rat-wound experiments were conducted to provide *in vivo* evidence of the interaction between DEG-HEP-COF and the wound environment, including the ability of COFs to adsorb and release drugs effectively. By evaluating the performance of DEG-HEP-COF carried antibiotic system in a relevant biological context, we aimed to explore the advantage of this kind of COF containing hetero-environmental pores and establish a comprehensive understanding of its potential in the application as drug delivery systems.

8. Overall, the idea of the "checkboard" alternation of hydrophilic and hydrophobic pockets is a good idea. In the current paper, this is not proven but assumed or speculated. The high amount of application distracts from the core message.

Thank Reviewer 3 very much for his/her insightful comments, we understand the importance of providing concrete evidence to support our claims. This manuscript from the perspective of crystallinity analyses, energy calculation, and double antibiotic loading application research of COFs proves the structure rationality of the proposed hetero-environmental pores. The obvious differences in the growth processes demonstrated the regulation of the branch chains on the structure and performance of COFs. Meanwhile, the STM with the atomic resolution was conducted to tentatively study the arrangement of the two side chain groups. From STM results, we can determine the location of TAPP and the structure of surface DEG-HEP-COF, but still cannot directly observe the side chain arrangement of DEG-HEP-COF. Hence the theoretical calculations were carried out to further explore the structure of DEG-HEP-COF. The calculation indicates the rationality of the alternant pore structure in the DEG-HEP-COF. We further synthesized the dialdehyde monomers with a single side chain (hydrophobic alkyl chain or hydrophilic ethylene oxide chain) to construct the sCOFs (Supplementary Schemes 5 and 6, Supplementary Figs. 26 and 27). The pore size distribution data infer the alkoxy chains have the directional orientation of a stacking. Significantly, the results of loading and releasing antibiotics of the DEG-HEP-COF indirectly prove the rationality of the proposed hetero-environmental pores. In the revised manuscript, we have made revisions to clearly indicate where assumptions are made and provide justifications for them. This will enhance the scientific rigour and credibility of our research.

Additionally, we understand his/her concern regarding the high amount of application examples included in the manuscript. After carefully reappraising the significance of the photocatalysis application part, although our intention is to illustrate the practical implications of the DEG-HEP-COF, we now realize that it might detract from the core message. In response to his/her suggestion, to enhance the clarity of the article's logic and highlight the key points, we have reconstructed the manuscript, deleted the photocatalytic part, and focused on drug delivery based on the unique pore structure of DEG-HEP-COF by specifically delving into the investigation of the relationship between the pore environment and the loading and releasing of antibiotics. Furthermore, the impact of pore environment on the optical and catalytic properties of materials will be extensively explored in the forthcoming research.

REVIEWER COMMENTS

Reviewer #1 (Remarks to the Author):

The manuscript is now significantly revised.

It could be published, by addressing one remark:

1: In the wound healing assessment experiment, A minor comment is that in Fig 5a, why does the residual colony plot on the agar broth plate show green while the background is black?

Reviewer #2 (Remarks to the Author):

The Authors made the required corrections, and the manuscript can be accepted for publication.

Reviewer #3 (Remarks to the Author):

The authors have made drastic modifications to the revised manuscript, and this has resulted in a much more focused document with a better experimental/theoretical corroboration of the research conclusions.

In my original review, I criticized the fact that the authors used their materials in three different applications and that some conclusions were rather speculative.

The photocatalytic applications and the growth mechanism analysis (speculative) have now been removed, focusing on ONE application (drug delivery). This is good.

The evidence for the checkboard alterations of hydrophobic/hydrophilic pockets is now much better corroborated. I understand that a 100% proof is never possible with the current available technologies, but authors have gone the extra mile to make the reasoning plausible and acceptable.

I still have a minor comment: authors have tried to convince me that only one peak in COFs is strong proof for a highly crystalline material. I am very active in this field and I know that this is not the case: One peak (eg(100) or (110)) in XRD only proofs a ordering of pores, but not crystallinity (see also templated mesoporous silicas). Nicely ordered imine COFs show typically several (4-8) peaks in the region 1-10 2theta, (there are dozens of examples in literature). Also, if the COFs are nicely AA stacked, they usually show a broad 001 reflection between 20-30 2theta. So I'd like to see that the authors tone down their claims on high crystallinity. A

For the rest, the authors have written a convincing rebuttal and the manuscript can be published after minor modifications.

Replies to Reviewers' Comments

Reviewer #1 (Remarks to the Author):

The manuscript is now significantly revised. It could be published, by addressing one remark: In the wound healing assessment experiment, A minor comment is that in Fig 5a, why does the residual colony plot on the agar broth plate show green while the background is black?

Thank Reviewer 1 very much. We are happy to know that he/she is satisfied with our revised manuscript. Regarding the color of the residual colony and background, we recolored the images to make the colonies more obvious for reviewers and readers' observation according to some reported articles. Per suggestion, Fig 5a has been replaced by the initial images.

Fig. 5a Residual bacteria colonies on agar broth plates separated from wounds with different treatments at 12 d.

Reviewer #2 (Remarks to the Author):

The Authors made the required corrections, and the manuscript can be accepted for publication.

Thank Reviewer 2 very much for his/her recommendation.

Reviewer #3 (Remarks to the Author):

The authors have made drastic modifications to the revised manuscript, and this has resulted in a much more focused document with a better experimental/theoretical corroboration of the research conclusions. In my original review, I criticized the fact that the authors used their materials in three different applications and that some conclusions were rather speculative. The photocatalytic applications and the growth mechanism analysis (speculative) have now been removed, focusing on ONE application (drug delivery). This is good. The evidence for the checkboard alterations of hydrophobic/hydrophilic pockets is now much better corroborated. I understand that a 100% proof is never possible with the current available technologies, but authors have gone the extra mile to make the reasoning plausible and acceptable.

Thank Reviewer 3 very much. We are happy to know that he/she is satisfied with our revised manuscript. The constructive criticism motivated us to delve deeper into the topic. This review has made a significant impact on our work.

I still have a minor comment: authors have tried to convince me that only one peak in COFs is strong proof for a highly crystalline material. I am very active in this field and I know that this is not the case: One peak (eg(100) or (110)) in XRD only proofs a ordering of pores, but not crystallinity (see also templated mesoporous silicas). Nicely ordered imine COFs show typically several (4-8) peaks in the region 1-10 2theta, (there are dozens of examples in literature). Also, if the COFs are nicely AA stacked, they usually show a broad 001 reflection between 20-30 2theta. So I'd like to see that the authors tone down their claims on high crystallinity.

For the rest, the authors have written a convincing rebuttal and the manuscript can be published after minor modifications.

Thank Reviewer 3 very much for his/her insightful comments, we fully agree with the comments and suggestions. Per suggestions, for the description of COF's crystallinity, we toned down the statements about COF's crystallinity and utilized the periodically ordered pore structure or (optimized) crystallinity in the revised manuscript. The changes are marked in green in the revised manuscript.

"The DEG-HEP-COFs with optimized crystallinity and Brunauer–Emmett–Teller (BET) specific surface area can be obtained under several different conditions."

"The high-resolution transmission electron microscopy (HR-TEM) image shows highly ordered lattice fringes, confirming the periodically ordered pore structure of DEG-HEP-COF."

"Thus, the amphiphilic DEG-HEP-CHO combining the alkyl chains and alkoxy chains exhibits a moderate reaction rate, which leads to the DEG-HEP-COF with ordered pore structure, uniform pore diameter distribution, and high BET specific surface area."

REVIEWERS' COMMENTS

Reviewer #1 (Remarks to the Author):

The authors have performed further experiments. They have also adequately addressed my comments and concerns.

Reviewer #3 (Remarks to the Author):

I am satisfied with the second revised version of this manuscript. It can now be published.